# Distinct Gene Expression Profiles in Colonic Organoids from Normotensive and the Spontaneously Hypertensive Rats

**DOI:** 10.3390/cells10061523

**Published:** 2021-06-17

**Authors:** Jing Li, Elaine M. Richards, Eileen M. Handberg, Carl J. Pepine, Mohan K. Raizada

**Affiliations:** 1Department of Physiology and Functional Genomics, University of Florida College of Medicine, Gainesville, FL 32610, USA; lijing1@ufl.edu (J.L.); esumners@ufl.edu (E.M.R.); 2Division of Cardiovascular Medicine, Department of Medicine, University of Florida College of Medicine, Gainesville, FL 32610, USA; Eileen.Handberg@medicine.ufl.edu (E.M.H.); Carl.Pepine@medicine.ufl.edu (C.J.P.)

**Keywords:** hypertension, colonic organoid, immune response, antigen presentation, butyrate

## Abstract

Hypertension is associated with gut bacterial dysbiosis and gut pathology in animal models and people. Butyrate-producing gut bacteria are decreased in hypertension. RNA-seq analysis of gut colonic organoids prepared from spontaneously hypertensive rats (SHR) and normotensive Wistar Kyoto (WKY) rats was used to test the hypothesis that impaired interactions between the gut microbiome and gut epithelium are involved and that these would be remediated with butyrate supplementation. Gene expressions in immune responses including antigen presentation and antiviral pathways were decreased in the gut epithelium of the SHR in organoids and confirmed in vivo; these deficits were corrected by butyrate supplementation. Deficits in gene expression driving epithelial proliferation and differentiation were also observed in SHR. These findings highlight the importance of aligned interactions of the gut microbiome and gut immune responses to blood pressure homeostasis.

## 1. Introduction

Increasing evidence implicates gut microbial dysbiosis in animal models of hypertension (HTN) [1,2,3,4] and in patients with elevated blood pressure (BP) [5,6,7]. Decreased butyrate-producing bacterial communities associated with HTN pathology was frequently observed [1,7]. Furthermore, butyrate supplementation reduced BP in hypertensive animals [8] and a clinical trial is underway to provide evidence for this concept (ClinicalTrials.gov Identifier: NCT04415333). Bacterial dysbiosis in hypertension is associated with significant gut pathology that include decreased tight junction proteins, increased gut wall thickness, shortened villi, reduced goblet cells and increased gut barrier leakiness [4,5,9]. This was also associated with profound effects in gut immune system [10]. Despite these changes, nothing is known about whether these changes in the gut are reflected in the epithelium. Furthermore, any link between changes in gut epithelial cells, its impaired immune system and blood pressure remains to be studied. Nonetheless, all available evidence suggests that the communication between the gut microbiota and gut epithelium could be key in BP homeostasis and impaired interplay could be associated with HTN.

Therefore, this study was undertaken to advance our understanding of the role of gut epithelium in HTN. We tested the hypothesis that altered gut microbiota changes epithelial functions epigenetically, resulting in impaired gut barrier function, immunity and overall metabolome in HTN. We elected to utilize three-dimensional (3D) organoids of proximal colons from normotensive Wistar Kyoto (WKY) rats and spontaneously hypertensive rats (SHR). The rationales were the following: (1) The colon harbors significantly more bacteria compared with other gastrointestinal segments and the proximal colon is close to the cecum, which is a site of high bacterial metabolism and fermentation [8,11,12,13]. Additionally, previously described HTN-related gut pathological features, dysbiosis and metabolic capacity are represented in the proximal colon [4,14]. (2) Intestinal epithelial cells respond to signals from commensal microbes or invading pathogens and adapt to alteration of the gut environment by modulating epithelial and mucosal function. These signals are bacteria-derived metabolites (e.g., short chain fatty acids, bile acid and tryptophan metabolites), bacterial components and bacteria themselves [11,13,15]. (3) Colonic 3D organoids are self-renewing structures embedded in extracellular matrix that maintain their colon-specific cell lineage, crypt-like proliferative zones and differentiate into all cell types found in the gut epithelium. Additionally, gut organoids contain polarized epithelial cells, tight junctions, secrete mucus and have remarkably stable epithelial phenotype and genotype [16]. Thus, 3D organoids provide a novel ex vivo system to investigate pathophysiological mechanisms. Our previous studies have established and characterized colonic organoids from WKY and SHR [17,18]. These cultures were employed in the present study to investigate potential epithelial–microbiota interactions and the effect of butyrate using RNA sequencing.

## 2. Materials and Methods

### 2.1. Animals and BP Measurement

Male fourteen-week-old WKY (Mean SBP: 132 ± 6 mmHg) and SHR (Mean SBP: 209 ± 5 mmHg) from Charles River were acclimated to housing conditions for 2 weeks. Systolic blood pressure (SBP) was measured by Tail-Cuff Plethysmography (Visitech Systems) with the plate temperature set at 35 °C. All studies were approved by the Institutional Animal Care and Use Committee of the University of Florida. Rats were maintained in accordance with the Animal Welfare Act and the Public Health Policy on Humane Care.

### 2.2. Colonic 3D Organoid Culture

Primary colonic crypts were isolated from proximal colons of WKY and SHR rats with gentle cell dissociation reagents (STEMCELL Technologies) including 2 mM EDTA for 50 min, which were grown and maintained as 3D-spheroid cultures in Matrigel (BD Biosciences) containing organoid growth medium (STEMCELL Technologies) as described previously [17,18].

### 2.3. RNA-Seq

Colonic organoids were cultured for 4 days and then treated with butyrate (Sigma-Aldrich, St. Louis, MO, USA) or without butyrate (control treatment) for 24 h in the following experiments. Total RNA was extracted from colonic organoids with RNeasy Plus Mini Kit (Qiagen, Hilden, Germany). cDNA was generated using SMART-Seq HT kit (Takara Bio, Kusatsu, Japan) and RNA-Seq libraries were constructed with Nextera DNA Flex Library Prep kit and Nextera DNA Unique Dual Indexes Set A (Illumina, San Diego, CA, USA) and sequenced on NovaSeq6000 (Illumina) at the University of Florida NextGen DNA Sequencing Core Facility. RNA seq analysis was performed using CLC genomics workbench (Qiagen).

### 2.4. Quantitative PCR (qPCR)

Total RNA was purified and reverse-transcribed using iScript Reverse Transcription Supermix (Bio-Rad, Hercules, CA, USA). Finally, qPCR (ABI Prism 7600, Foster City, CA, USA) was performed with Taqman universal PCR master mix and specific probes (*Alpi*, *RT1-CE5*, *RT-Ba*, *Cd74*, *Ifi44l*, *Ifit2*, *Cbr1*, *Cbr3*, and *Btnl7*). *Gapdh* was used to normalize the expression of these genes and fold change was calculated relative to WKY using the ΔΔCt method.

### 2.5. Western Blot

Proximal colons and organoids from WKY and SHR were homogenized in 2% SDS-Tris buffer (pH = 7.5). Protein samples were separated on 12% TGX precast gels and transferred to Polyvinylidene fluoride membranes (Bio-Rad). Membranes were incubated with RT1-A (1:800, Invitrogen) and Beta (β)-actin (1:2000, Abcam, Cambridge, UK) antibodies followed by anti-rabbit and anti-mouse IRDY 680RD secondary antibodies (Li-Cor Biosciences, Lincoln, NE, USA). Protein bands were detected using the Odyssey infrared imaging system.

### 2.6. Immunofluorescence Imaging

Proximal colons from WKY and SHR rats were excised and fixed in 4% paraformaldehyde in PBS for 48 h, embedded in paraffin and sectioned at 5 µm. Briefly, after deparaffinization and rehydration, slides were incubated in antigen retrieval solution (10 mM sodium citrate, pH 6.0) at 92 °C for 40 min. Sections were permeabilized (0.5% Triton X-100 in PBS), blocked (5% bovine serum albumin and 0.3% Triton X-100 in PBS) and incubated with Epcam (1:500, Invitrogen, Carlsbad, CA, USA) and RT1-A (1:200, Invitrogen) antibodies overnight. Finally, the sections were stained with DAPI, anti-mouse Alexa Fluro 594 and anti-rabbit Alexa Fluro 488 secondary antibodies (Thermo Fisher Scientific, Waltham, MA, USA) and then imaged with a confocal microscope (Olympus IX81, Shinjuku, Japan), 20× objective, Scale bar: 50 µM.

Colocalization analysis of immunofluorescence intensity was performed as previously described [19]. The mean intensity of immunofluorescence of RT1.A and Epcam was quantified and normalized to the area with Image J software. Colocalization values were calculated from nine representative images using the colocalization Coloc 2 (Pearson’s coefficient).

## 3. Results

### 3.1. Differential Transcriptional Program in Gut Organoids of the WKY and SHR

Our first objective was to advance the understanding of dysfunctional gut epithelial-microbiota communication reflected in gut pathology and gut bacterial dysbiosis in HTN. Thus, we performed high-throughput RNA sequencing analysis of genome-wide mRNA expression in 3D organoids from 14-week-old WKY rats and SHR. We identified 539 genes that were significantly changed between the two groups of rats; 333 were downregulated and 206 upregulated (FDR-corrected *p* < 0.05, absolute fold change ≥ 2) in the SHR (Figure 1A). Gene set enrichment analysis (GSEA) of these differentially expressed genes (DEGs) disclosed that gene sets for immune response were significantly downregulated in colonic organoids from SHR (Figure 1B). They included gene sets associated with antigen presentation (AP), with normalized enrichment score (NES) = −2.1577, adjusted *p* = 0.016) and immunoregulation between a lymphoid and a non-lymphoid cell (NES = −2.1259, adjusted *p* = 0.015) (Figure 1B). In contrast, gene sets for mitochondria metabolic activities in colonic organoids, e.g., inner mitochondrial membrane protein complex (NES = 1.8839, adjusted *p* = 0.022) and oxidation-reduction process (NES = 1.3979, adjusted *p* = 0.036) were enriched in SHR organoids (Figure 1B).

Heatmap analysis further elaborated altered levels of specific genes between WKY and SHR under major annotated pathways (Figure 1C). An interesting set of mRNAs for which expression significantly decreased were the AP cascade (e.g., *RT1B*, *Cd74* and *Tap1/2*) for the major histocompatibility complex (MHC) class I and II proteins. Similarly, mRNAs of genes associated with immune response were decreased in the SHR. Most notable were *Ccl5* (C-C Motif Chemokine Ligand 5), *Nlrc5* (NLR Family CARD Domain Containing 5), which is a key transcriptional regulator of MHC class 1, *Lcn2* (lipocalin2) and *Btnls* (Butyrophilins, major epithelial determinants of tissue-associated γδ T cell compartments). Furthermore, mRNAs linked to the defense response to viruses, particularly *Ifit2* and *Ifit3* (Interferon induced protein with tetratricopeptide repeats), that inhibit viral replication and that has potent antiviral activity against HIV-1, such as *Mx2* (Interferon-induced GTP-binding protein), were decreased.

Decrease in mRNA expression for *Alpi* (alkaline phosphatase, intestinal) in the SHR and increases of *Dpt* (Dermatopontin) and *Rerg* (RAS such as estrogen regulated growth inhibitor) were observed. These genes are shown to be involved in cell proliferation, migration and differentiation, while *Alpi* is involved in detoxification of LPS.

Other genes for which their expression could impact epithelium–microbiota dysfunction in the SHR include those that were increased: *Mthfr* (Methylenetetrahydrofolate reductase, involved in folate metabolism and homocysteine synthesis); *St6galnac2* (N-Acetylgalactosaminide Alpha-2,6-Sialyltransferase 2), an enzyme that regulates sialylation of glycoprotein mucin, actin, etc.; *Antptl8* (Angiopoietin-like Protein 8), involved in regulation of triglyceride and HDL levels; *Ephx2* (Epoxide Hydrolase 2), regulates lipid epoxides such as epoxyeicosatrienoic acid (EETs) (increased EETs are associated with cardiovascular disease); *Hmgb2* (high-mobility group protein 2), that has antibacterial, inflammatory and oxidation-reduction activity in GI tract; *Cbr1* (Carbonyl reductase 1), an NADPH-dependent oxidoreductase involved in oxidative stress and apoptosis; *Ndufa10/1* (NADH dehydrogenase [ubiquinone] 1 alpha subcomplex 10/1), a subunit of the mitochondrial membrane respiratory chain. Furthermore, those that were decreased also impact epithelium–microbiota dysfunction: G-protein signaling-linked genes such as *Rgs14* (Regulator of G-protein signaling 14), that integrates G protein and Ras/ERK signaling pathways to attenuate G-protein signaling; *Rasa4* (Ras GTPase), that also suppresses ERK signaling in response to calcium influx; *Adrb2* (Adrenoceptor Beta 2), involved in gut inflammatory responses). mRNAs for three genes in ubiquitination process, *Lrsam1* (Leucine rich repeat and sterile alpha motif containing 1), *Ube3d* (Ubiquitin protein ligase E3D) and *Usp18* (Ubiquitin specific peptidase 18) were decreased in the SHR. Taken together, these observations suggest that AP system, immune response and genes involved in viral defense mechanisms are significantly compromised in the SHR gut epithelium; these impacts signaling, metabolism and ultimately proliferation and differentiation.

Next, we used qPCR to validate the RNA-seq data of key genes. Figure 2A shows that mRNA levels for *Alpi*, *RT1-CE5*, *RT1-Ba*, *Cd74*, *Ifi44I* and *Ifit2* were significantly decreased while mRNAs for *Cbr1*, *Cbr3* and *Btnl7* were increased in SHR organoids consistent with the RNA-Seq data. mRNAs of two key genes, *RT1-CE5* and *RT1-Ba*, of AP cascade were similarly decreased in proximal colon (Figure 2B). Furthermore, the RT1-A protein was also decreased in both proximal colon and organoids of the SHR (Figure 2C and Appendix A).

We further examined the expression and localization of RT1-A, a major MHC class 1 protein in proximal colon sections from WKY and SHR rats. RT1-A was preferentially and abundantly expressed in apical membranes of colons from both groups of rats (Figure 3A and Appendix A). However, the fluorescence intensity was significantly greater in WKY colon sections than SHR sections (WKY 20.99 ± 4.402 vs. SHR 10.31 ± 1.765, *n* = 9, *p* = 0.0244; Figure 3B). This is consistent with RT1-A mRNA and protein expression (Figure 2C). The RT1A was primarily colocalized with Epcam, which is a marker for epithelial cells. Interestingly, the colocalization coefficient was significantly decreased in the SHR colonic epithelium without any change in Epcam expression between the two groups (WKY 0.4467 ± 0.04738 vs. SHR 0.2967 ± 0.03436, *n* = 9, *p* = 0.0208; Figure 3C,D). As a result, the RT1.A to Epcam fluorescence ratio of WKY was 1.6-fold higher than SHR (WKY 33.33 ± 3.579 vs. SHR 20.27 ± 2.473, *n* = 9, *p* = 0.0085; Figure 3E). These data further confirm that the antigen presentation cascade is compromised in the SHR.

### 3.2. Abundance of Bacterial Taxa and Expression Profile of Antigen Presentation Genes in WKY and SHR

Gut microbiota plays an important role in the regulation of AP-related gene expression and altered microbiota–epithelium communication resulting in impaired immune and neural signaling and gut pathophysiology associated with many chronic diseases, including graft-versus-host and Crohn’s disease [20,21]. Our previously published data on microbial communities in WKY and SHR had identified the enrichment of five bacterial taxa (*Allobaculum*, *Bacteroides*, *Bifdobacterium*, *Blautia* and *Gordonibacter*) in WKY rat and one taxa (*Turicibacter)* in the SHR microbiomes [1]. Interestingly, we found that the increased expression of most AP-related genes in the WKY rat gut epithelium coincided with increases in these taxa.

### 3.3. Butyrate Treatment Restores Gene Expression for AP in SHR Organoids

*Bifdobacterium*, *Allobaculum* and *Blautia* are all butyrate-producing bacteria for which levels are depleted in SHR [1]. Decreased plasma butyrate has been reported in HTN [22]. Furthermore, butyrate supplementation alleviates HTN pathophysiology [8]. This finding led us to determine if butyrate treatment would rescue impaired AP-associated gene expression in the SHR. Reads per kilobase per million (RPKM) of MHC class I (*RT1-CE3*, *RT1-CE5*, *RT1-CE16*, *RT1-A2*, *RT1-CE16*, *Tap1/2* and *RT1-N2*) and II (*RT1-Ba*, *RT1-Bb*, *RT-Db1* and *Cd74*) genes are significantly decreased in SHR organoids compared to WKY organoids (Figure 2B and Figure 4); butyrate treatment significantly elevated the expression of their mRNAs to close to (e.g., *RT1-Ba*, *RT1-CE3* and *RT1-CE5*) or higher than (e.g., *RT1-Bb*, *RT-Db1* and *RT1-N2*) the baseline expression in WKY (Figure 4). Interestingly, butyrate increased these genes in WKY organoids as well (Figure 4). Basal expression of another notable gene for MHC complexes, *Ciita* (Class II Major Histocompatibility Complex Transactivator), was not significantly different in WKY vs. SHR organoids. However, butyrate treatment resulted in a 2-fold greater increase in *Ciita* mRNA in WKY rat than in SHR organoids (Figure 4). CIITA plays a key role in boosting the transcriptional activity of MHC class 1 and 2 genes thus enhancing the ability of the immune system to combat foreign proteins; butyrate enhanced this effect.

## 4. Discussion

The most significant finding of this study is that the expression of genes related to AP in the MHC molecular cascade is significantly decreased, compromising the mounting of immune and other defense responses in the SHR (Figure 5). This could result in altered epithelial cell proliferation, differentiation and communication with microbiota in part, by depletion of butyrate-producing bacterial communities in this model of hypertension (Figure 5). Supplementation with butyrate seems to overcome deficiencies in butyrate-producing bacteria, rebalances AP-related gene expression, improves gut pathology and leakiness and lowers blood pressure ([5], Figure 5).

Intestinal epithelial cells (IECs) have antigen processing and presentation machinery and express MHC class I and MHC class II molecules. Thus, IECs are able to integrate signals from the microbiota in the gut lumen in a complex interplay of bacteria and mucosal immune system. For example, MHC class I molecules are critical in the transmission of bacterial-derived lumen-generated signals to the submucosa that influences epithelial cell function, proliferation and differentiation, which ultimately regulates overall gut immune status and its defense responses. For example, the RT1 group of genes in MHC class I are linked to T cell receptor signaling that works in combination with genes for immune regulatory molecules such as IFNγ, PA28, HSP70 and TAP1/2 [23,24]. Decreased expression of these RT1 genes, particularly Ba, CE3, CD74 and the Bb group, in the SHR would compromise AP signaling resulting in ineffective immune responses to gut dysbiosis and the altered luminal environment in hypertension. This view is consistent with decreases in other immune response genes in the SHR. A few most relevant genes are also described here. Btn (Butyrophilin) or Btlns (butyrophilin)-like molecules, many of which are located within the MHC, are implicated in chronic inflammatory diseases [25]. Polymorphisms of BTNLs and sometimes their deficits have been associated with chronic kidney disease, hypertension and diabetes [26,27,28,29,30]; Nlrc5, a pattern recognition receptor in the MHC I class, regulates NF-κB, type 1 interferon activities and JAK/STAT3 signaling in immune response pathways. *Nlrc5* deficiency is associated with neointimal hyperplasia, aortic smooth muscle proliferation, chronic inflammation and diabetic retinopathy [31]; Therefore, its decrease is consistent with gut pathophysiology in hypertension [4]. Lcn2 (Lipocalin 2) produces neutrophil gelatinase-associated lipocalin (NGAL) protein that sequesters iron-containing siderophores of the enterochelin type that are sequestered by bacteria that are dependent upon these specific siderophores to boost their iron uptake [32]. Binding of NGAL to these bacterial siderophores chelates iron, which controls growth of these bacteria and the innate immune responses with respect to the infections with them [33]. Thus, decreased expression of *Lcn2*/NGAL would favor bacteria that rely on enterochelin-based iron uptake (many of which are human pathogens) in the SHR gut, resulting in a decrease in beneficial bacteria but an increase in pathogenic bacteria. The decreased expression of genes relevant to ubiquitination (*Lrsam*, *Ube3d* and *Usp18*) is consistent with the established role of ubiquitination and autophagy in the MHC class I antigen presentation pathway [34]. In addition, ubiquitination signaling molecules, such as NF-κB, interferon and interferon-induced proteins, are all linked with the regulatory events associated with immunity, apoptosis and epithelial cell proliferation [35]. Finally, increased expression of *RT1-S2*, involved in antigen presentation in MHC class I, is intriguing in view of the evidence that most of the RT1 genes are decreased (Figure 1B). This gene is associated with blood pressure quantitative trait locus (QTL) in the rat [36] and thus may possess unrecognized significance.

Another group of genes for which its expression could have a major impact on hypertension are those associated with the defense response to virus. This is particularly interesting in view of the increasing evidence for the role to the virome, especially bacteriophages in gut microbiome-associated chronic diseases [37]. Interestingly, cluster analysis has demonstrated a significant alteration in the gut virome, particularly of bacteriophages, in patients with hypertension [38]. Our data in this study shows that many genes relevant to defense responses to virus were decreased (Figure 1B). This, one could postulate, may result in an impaired response to the blooming of pathogenic viral communities linked to hypertension. Some genes directly relevant to viral replication and infection in the SHR are the following: *Ifi44l* (Interferon-induced protein 44-like), its increased expression is linked to inhibition of viral replication while its loss has been associated with severe infection with respiratory syncytial virus [39,40]; *Mx2* (Mx dynamin such as GTPase 2) is an antiviral group of effector proteins of the types I and II interferon system and involved is in inhibition of early steps of viral replication [41]; similarly, *Rsad2* (Radical-S-adenosylmethionine superfamily enzyme also known as viperin protein) exhibits antiviral activity against a vast range of viruses [42]; finally, *Ifit* proteins (Interferon-induced proteins with tetratricopeptide repeats) are induced by interferon and mount a robust antiviral response [43]. Decreases in *Ifit2* and *Ifit3* in the SHR would result in decreased protection from viruses in the gut. Interestingly and counterintuitively, based on *Ifit2* and *Ifit3*, the expression of *Ifit1* was increased. However, based on its distinct mechanism of action, it is possible that *Ifit1* is involved in the replication of beneficial viruses. Ifit1 binds to the capped 5′ ends of viral mRNA only if the cap does not have 2′O-methylation to inhibit translation of the RNA [44]. Thus, its increased expression provides protection against infection from Hepatitis B, HPV, HCV, etc., which apparently possesses little relevance to hypertension.

It is pertinent to mention that the lack of robust antiviral responsiveness of the SHR epithelium could have implications beyond hypertension. For example, the COVID-19 pandemic has revealed that hypertensive people are highly vulnerable to severe outcomes of COVID-19. This view is supported by evidence that (a) hypertension is the most prevalent comorbidity associated with unfavorable outcomes in patients with COVID-19 [45]; (b) increasing evidence implicates involvement of the gut epithelium in the establishment and overall progression of COVID-19 [46,47]; (c) gut epithelium expresses one of the highest levels of ACE2, the receptor for SARS-CoV-2 and its partners required for infection [48]; (d) butyrate-producing bacteria are depleted in both hypertension and COVID-19 and butyrate supplementation appears beneficial in COVID-19, hypertension pathophysiology and gene expression in the antigen presentation pathway (present study).

We believe that it worthwhile to discuss several other genes because of their relevance to the interactions between gut epithelium and microbiota and, thus, their potential role in overall blood pressure homeostasis. They are the following: ALPI is an enzyme of the gut mucosal defense system that detoxifies lipopolysaccharides (LPS) and other bacterial products and prevents their translocation across the mucosa [49]. Decrease in *Alpi* in the SHR gut epithelium would profoundly compromise detoxification of LPS, etc., and exacerbate inflammatory responses to LPS. Decreased mRNA for *Alpi* in SHR gut organoids is consistent with our previous findings [14]. Furthermore, they advance the concept that this epigenetic change is maintained in vitro and persistent interaction with the gut microbiome is not essential. This enzyme also regulates gut wall barrier function, its decrease is associated with dysbiosis and the diffusion of bacterial, bacteriophage, metabolites and even populations of live bacteria into the blood. Furthermore, it is implicated in dysfunctional gut–brain, gut–liver and gut–lung communication in many pathophysiological states [50]. *Mthfr* gene product is associated with folate, homocysteine and methionine levels and metabolism [51]. Certain SNPs and polymorphisms of *Mthfr* are present in 24–87% of high blood pressure cases and 40% of CVD cases [52,53]. EPHX2 is a key enzyme in metabolism of epoxides and its inhibition increases EETs while it simultaneously decreases blood pressure and other cardiopulmonary pathophysiology. Inhibitors of epoxide hydrolase produce beneficial effects in intestine-related inflammatory diseases involving dysbiosis including IBD [54,55]. Finally, cholecystokinin (CCK) is responsible for the digestion of fats and proteins and the regulation of autonomic nervous system [56]. CCK activates vagal pathways that send anti-inflammatory signals and lower blood pressure [57,58]. Thus, decreased epithelial CCK in the SHR is likely to dampen vagal activity and help maintain high blood pressure.

Collectively, these observations suggest that hypertensive signals in the SHR initiate and establish the gut luminal environment to influence bacterial genetic pathways conducive to impaired immune responses and dampened responses for fighting viral infection. These changes are accentuated by supportive changes in calcium and G protein-coupled signaling and ubiquitination ultimately resulting in impaired epithelial function, proliferation and differentiation compromising gut wall functions.

The study has a limitation: We have primarily used RNA-seq and limited gene validation by qPCR to establish altered antigen presentation and related genes in the SHR. Functional data on these genes needs to be obtained to support their physiological relevance in gut immunity and blood pressure control. Thus, further studies are needed to determine if bacterial taxa from WKY rats would rebalance gene profiles in SHR colonic organoids in the same way that it can decrease blood pressure in the SHR [59].

## Figures and Tables

**Figure 1 cells-10-01523-f001:**
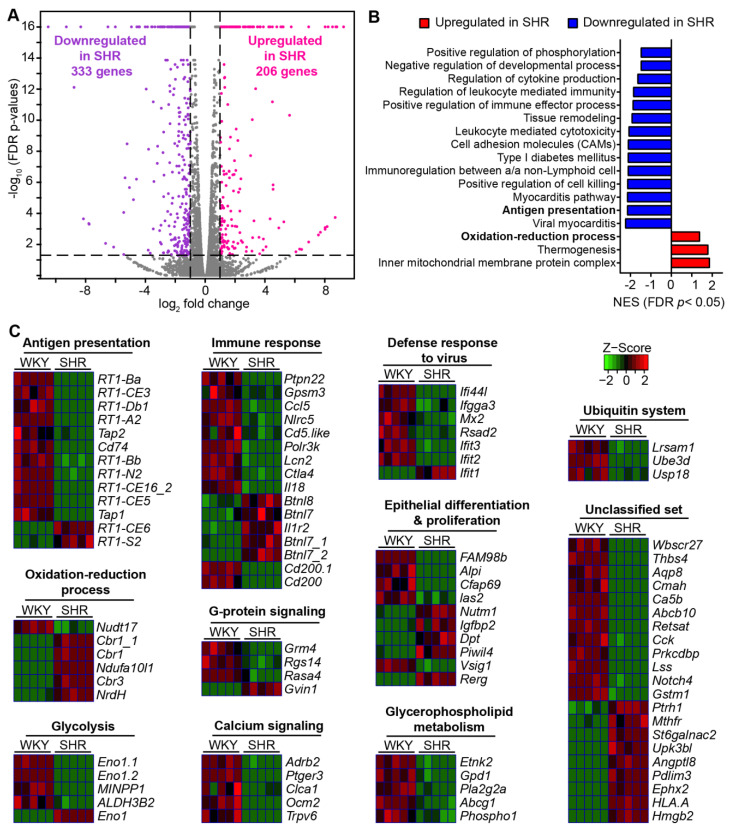
Comparison of the transcriptional program of colonic organoids of WKY and SHR. (**A**) Volcano plot showing differentially expressed genes (DEGs) comparing the colonic organoids of WKY with SHR (FDR-corrected *p* < 0.05, absolute fold change ≥ 2, purple dots show downregulated genes in SHR, pink dots show upregulated genes in SHR). Gray plots indicate that the genes were not differentially expressed (*n* = 5 per group). (**B**) Hallmark pathways significantly enriched in colonic organoids of SHR compared with WKY (Adjusted *p* < 0.05). NES: normalized enrichment score. (**C**) Heat map of DEGs in the annotated pathways, FDR-corrected *p* < 0.05, absolute fold change ≥ 2.

**Figure 2 cells-10-01523-f002:**
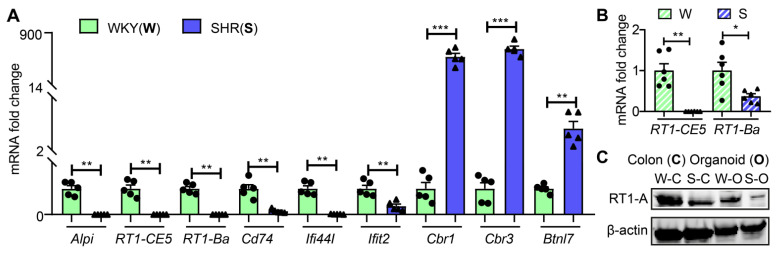
Validation of RNA-seq data ex vivo and in vivo. (**A**) mRNA expression of representative genes in colonic organoids from WKY and SHR rats by qPCR (*n* = 5 per group). (**B**) mRNA levels of *RT1-CE5* and *RT1-Ba* by qPCR in the proximal colons of SHR and WKY rats (*n* = 6 per group). Fold changes relative to WKY group. GADPH expression was used for normalization. Values are means ± SEM. * *p* < 0.05, ** *p* < 0.01 and *** *p* < 0.001, unpaired *t* test or Mann–Whitney U test. (**C**) Protein levels of MHC II class gene, RT1-A in organoids and proximal colons of WKY and SHR rats by Western blot. β-actin was used as a reference protein.

**Figure 3 cells-10-01523-f003:**
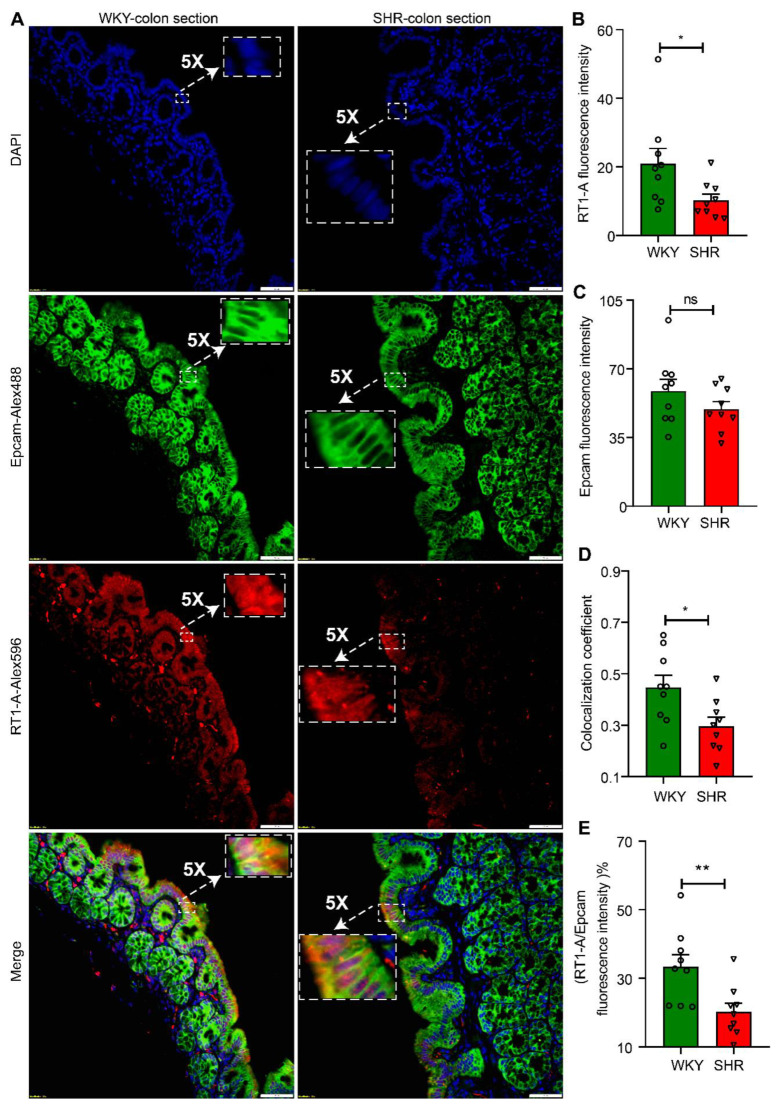
Downregulation of RT1-A expression in colonic epithelium of SHR rats. (**A**) Fluorescent immunohistochemistry staining of RT1-A and Epcam in proximal colon sections of WKY and SHR rats. Epcam is a marker of epithelium (20× objective, scale bar: 50 μm). (**B**–**E**) Quantitative analysis of RT1-A and Epcam expression and colocalization by Image J software, (**B**) RT1-A fluorescence density, (**C**) Epcam fluorescence intensity, (**D**) the extent of colocalization of RT1-A with Epcam, and (**E**) the percentage of RT1.A to Epcam fluorescence intensity, *n* = 9/group. Values are means ± SEM. nsp > 0.05, * *p* < 0.05, and ** *p* < 0.01, unpaired *t* test or Mann–Whitney U test.

**Figure 4 cells-10-01523-f004:**
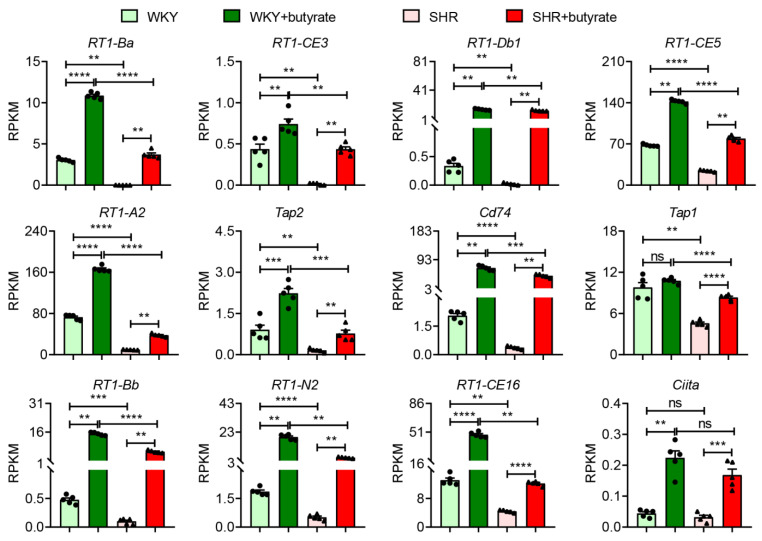
Butyrate rescues impaired expression of genes for antigen presentation in hypertensive gut organoids. Reads per kilobase per million (RPKM) of the genes involved in antigen presentation in organoids of WKY and SHR rats by RNA-seq with and without 3 mM butyrate treatment, *n* = 5/group. Values are means ± SEM. ns *p* > 0.05, ** *p* < 0.01, *** *p* < 0.001 and **** *p* < 0.0001, unpaired *t* test or Mann–Whitney U test.

**Figure 5 cells-10-01523-f005:**
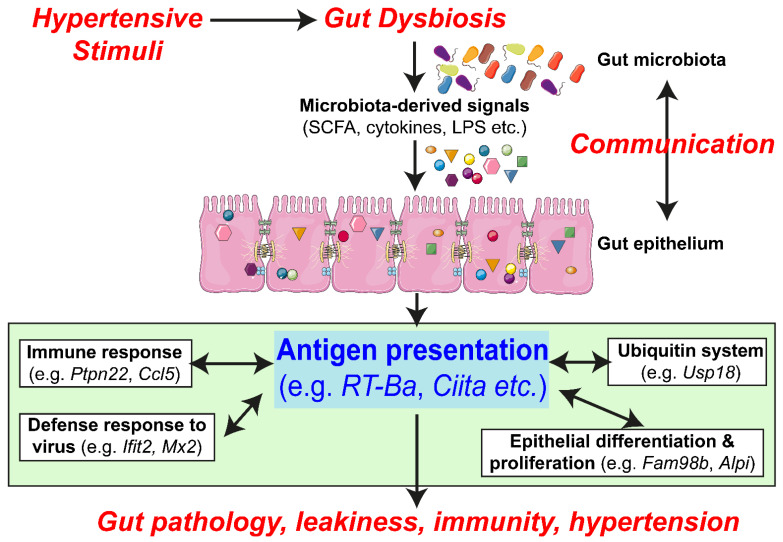
Impaired epithelial–microbiota communication in the SHR: proposed hypothesis. Hypertensive stimuli (genetic predisposition, epigenetic influences, drugs, etc.) by a still unexplored mechanism causes dysbiosis resulting in unique bacterial communities in hypertensive animals. This alters microbiota-derived signaling metabolites and molecules that effect AP genes and the cascade of related genes, e.g., immune response (*Ptpn22* and *Ccl5*), defense response to virus (*Ifit2* and *Mx2*), Ubiquitin system (*Usp18*) and epithelial cell proliferation and differentiation (*Fam98b* and *Alpi*). This, we propose, would initiate a series of signaling events resulting in increased gut leakiness, gut wall pathology, compromised gut and peripheral immunity. All this would induce a dysfunctional gut-brain axis and perpetuate the development and establishment of hypertension.

## Data Availability

Not applicable.

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
