# Peer review of "Distinct Gene Expression Profiles in Colonic Organoids from Normotensive and the Spontaneously Hypertensive Rats"

_cells, 2021, doi:10.3390/cells10061523_

Round 1

Reviewer 1 Report

This article has scientific quality and presents some novelty, filling in fact a void in scientific knowledge concerning the gut microbiome and gut immune responses to blood pressure homeostasis. Nevertheless, some minor considerations have to be clarified or fixed.

The title should be adequate to the content of the article. In the present study, we have normotensive rats and spontaneously hypertensive rats, also. Please consider to revise it.

Please consider adding a small subchapter, no more than 3-4 lines, about, intestinal epithelial cells (IECs) and immune responses to blood pressure homeostasis (What we known, already ?)

Line 32: «Dysbiosis in hypertension is associated……». Βacterial dysbiosis ?

Line 135:  It is wise to use the word  «stains»  when we study gut pathology and bacterial dysbiosis?

A «limitation» section should be structured.

Author Response

We thank the reviewer for the constructive comments. We have revised the manuscript accordingly and our point-by-point responses are as follows:

Reviewer #1:

The title should be adequate to the content of the article. In the present study, we have normotensive rats and spontaneously hypertensive rats, also. Please consider to revise it.

Response:

We agree and have revised the title to be more reflective of our data and conclusions.

Please consider adding a small subchapter, no more than 3-4 lines, about, intestinal epithelial cells (IECs) and immune responses to blood pressure homeostasis (What we known, already ?)

Response:

We understand the reviewer’s point. Unfortunately, we are not aware of any study that addresses the role of intestinal epithelial cells (IEC) and immune responses in blood pressure homeostasis other than our own studies. Therefore, we have summarized our published data and have included several lines in the “Introduction” mentioning the role of IEC in immune responses in general terms.

Line 32: «Dysbiosis in hypertension is associated……». Βacterial dysbiosis ?

Response: Corrected as suggested.

Line 135:  It is wise to use the word «stains»  when we study gut pathology and bacterial dysbiosis?

Response: Corrected from “strains” to “groups” in the text where appropriate. Also changes staining to expression when describing fluorescent microscopy results.

A «limitation» section should be structured.

Response:

This has been added at the end of the “Discussion” section.

Reviewer 2 Report

Li et al used colonic organoids derived from normotensive and spontaneously hypertensive rats to determine the role of butyrate in gut microbial dysbiosis associated with hypertension. RNA-Seq revealed distinct gene expression profile in the organoids from hypertensive rats and some of the differences between organoids from normotensive and hypertensive rats appeared to be partly abolished by butyrate supplementation, although a butyrate effect on the control organoid was also observed. The majority of the results presented in this manuscript were gene expression data, which seemed somewhat superficial to me when there were no functional data to support/confirm physiological relevance. Along this notion, there seemed to be insufficient data to test the hypothesis set out in Lines 39-41. I also had the impression that the authors were making claims/notions without supporting data from this study, e.g. what was the basis for “impaired epithelial-bacterial communications” (title)? Which dataset supported the following notion in the Discussion: “Supplementation with butyrate… improves gut pathology and leakiness and lowers blood pressure” (Lines 299-300)? How about “These observations clearly indicate unique bidirectional gut microbiota-epithelium signaling in hypertensive compared with the normotensive situation” (Lines 416-417)? Finally I found a lack of data that linked gut microbiota, intestinal epithelial functions and blood pressure a major pitfall of this study. My other specific comments are as follows:

  1. Were the same animal cohorts used in the organoid experiments and the gut microbiome analysis? In Lines 216-219 it sounded like the microbiome data were from a previous study (and thus the assumption that these might be different animals).
  2. Lines 115-116: “V4… of 16 RNA gene” was incorrect.

Author Response

We thank the reviewer for the constructive comments. We have revised the manuscript accordingly and our point-by-point responses are as follows:

Reviewer #2:

The majority of the results presented in this manuscript were gene expression data, which seemed somewhat superficial to me when there were no functional data to support/confirm physiological relevance. Along this notion, there seemed to be insufficient data to test the hypothesis set out in Lines 39-41. I also had the impression that the authors were making claims/notions without supporting data from this study, e.g. what was the basis for “impaired epithelial-bacterial communications” (title)? Which dataset supported the following notion in the Discussion: “Supplementation with butyrate… improves gut pathology and leakiness and lowers blood pressure” (Lines 299-300)? How about “These observations clearly indicate unique bidirectional gut microbiota-epithelium signaling in hypertensive compared with the normotensive situation” (Lines 416-417)? Finally I found a lack of data that linked gut microbiota, intestinal epithelial functions and blood pressure a major pitfall of this study.

Response:

  • We acknowledge that functional data would be important to confirm physiological relevance of these changes in genes expression and their correlation with microbial species. The functional experiments are complex and require extensive planning especially in the context of identifying specific bacterial species, their availability, validating their interactions with organoids and then validating in the physiological setting of whole animals. Therefore, we believe that it would be a separate study and is at the organizational stages. In our opinion, lack of this data does not diminish the significance of our findings in this paper. We have included this issue in the limitation section in the “discussion” to acknowledge this reviewers concern.
  • We have modified the hypothesis and revised the title per the reviewer’s recommendation.
  • We have included reference #5 for changes in gut pathology, blood pressure etc. by butyrate treatment in a rodent model of hypertension.
  • The sentence related to “bidirectional gut microbiota ---” is premature and has been deleted.
  • We have acknowledged lack of functional data above; such studies are being planned and will be undertaken soon.
  1. Were the same animal cohorts used in the organoid experiments and the gut microbiome analysis? In Lines 216-219 it sounded like the microbiome data were from a previous study (and thus the assumption that these might be different animals).

Response: The reviewer is correct. We used different set of animals for microbiota and organoid experiments. This has now ben stated at appropriate place in the Methods.

  1. Lines 115-116: “V4… of 16 RNA gene” was incorrect.

Response: corrected as “V4 to V5 variable region of 16S ribosomal DNA” in the text.

Round 2

Reviewer 2 Report

The revision and response to comments are noted and greatly appreciated. While the manuscript has been improved, a major experimental flaw was performing correlation analysis using microbiome and gene expression data from different animal cohorts. It is inappropriate to assume that animals of different cohorts, albeit receiving the exact same treatment, to have the same gut microbiota.

Author Response

We acknowledge the reviewer point that the data from the same animal cohort would have been ideal. We have added this under limitations and have also indicated high reproducibility of gut microbial dysbiosis data in the SHR by our group and others over the last several years in multiple separate publications.  

Round 3

Reviewer 2 Report

Performing correlation analysis using microbiome data from a separate animal cohort remained a serious flaw in the experimental design that could not be overlooked.

Author Response

We acknowledge the reviewer point and have deleted the data (Figure 5) and mention of correlation throughout the manuscript as recommended by the editor. 
